

# X-ray diffraction and theoretical study of the transition 2H-3R polytypes in $Nb_{1+x}Se_2$ $(0 < x < 0.1)$

Mohamed Sidoumou[1], Soumia Merazka[2], Adrian Gómez-Herrero[3], Mohammed Kars[2] and Roisnel Thierry[4]

[1] Département de Physique, Laboratoire de Physique Théorique et Interaction Rayonnement Matière, Université de Blida-1, Soumaa Blida, Algérie
[2] Faculté de Chimie, Laboratoire Sciences des Matériaux, Université Houari-Boumedienne, Algérie, El-Alia Bab-Ezzouar, Algérie
[3] Centro de Microscopia Electrónica, Universidad Complutense de Madrid, Madrid, Spain
[4] CNRS, ISCR (Institut des Sciences Chimiques de Rennes), Université de Rennes 1, Rennes, France

## ABSTRACT

Single crystals of 2H and 3R niobium diselenide were grown by a chemical transport reaction. The current determinations by single crystals X-ray diffraction reveal a non-stoichiometric composition. The structures are built from Se—Nb—Se slabs with Nb in trigonal prismatic coordination whereas the extra or additional Nb atoms are located in the octahedral holes between the slabs giving rise to the formula 2H and $3R$-$Nb_{1+x}Se_2$ with $0.07 < x < 0.118$. In particular, vacancy and Nb-Nb interactions may play an important role on the non-stoichiometry and the stacking mode in $NbSe_2$. By increasing the number of additional Nb atoms in the pure 2H-$NbSe_2$, a transition 2H to 3R polytype should occur in order to minimize the $Nb_{layer}$—$Nb_{extra}$—$Nb_{layer}$ repulsions between these adjacent slabs. The theoretical study shows that both 2H and 3R-$Nb_{1+x}Se_2$ are thermodynamically stable in the range $0 < x < 0.1$.

## INTRODUCTION

$NbSe_2$ belongs to the transition metal dichalcogenides TMDC's with chemical formula $TX_2$ (T = Nb, Ta; X = S, Se), which have recently renewed interest because of their quasi 2D nature similar to graphene making them very promising in novel electronic devices applications (*Vogel & Robinson, 2015*). The system $NbSe_2$ has been the subject of many investigations since it exhibits incommensurate charge density waves (CDW) and superconductivity phenomenon above 4K (*Wilson, Disalvo & Mahajan, 1975*).

The structure consists in hexagonally arranged Se-Nb-Se sandwiches with metal atoms (Nb) that are located between two layers of chalcogen atoms (Se) in a trigonal prismatic coordination. The bonding within each sandwich is covalent while the bonding among sandwiches themselves is a weak Van-der Waals type. Although the stacking along the c axis gives rise to several polytypes usually described as 1T–$NbSe_2$, 2H- $NbSe_2$, 3R–$NbSe_2$

Corresponding author
Mohammed Kars, mkarsdz@yahoo.fr

and 4H- $NbSe_2$ (*Kalikhman & Umanskii, 1973*; *Brown & Beerntsen, 1965*), it seems that only the 2H and 3R are frequently obtained in practice.

Until date, there are many reports on polymorphism in the TMDC's by tuning synthesis temperature such as the high temperature monoclinic $MoTe_2$ (*Brown, 1966*), the layered telluride $Mo_{1-x}Nb_xTe_2$ (*Ikeura et al., 2015*), polytypism in $TiS_2$ which was first observed by *Tronc & Huber (1973)* and *Legendre et al. (1983)*, the highly polymorphic $TaSe_2$ (*Brown & Beerntsen, 1965*; *Huisman, Kadijk & Jellinek, 1970*) and the polymorph $TaSe_{2-x}Te_x$ (*Luo et al., 2015*).

This structural anisotropy enables the intercalation of species to form stoichiometric and non-stoichiometric compounds with different guests. A number of authors (*Huisman, Kadijk & Jellinek, 1970*) and references therein) reported the existence of the Nb self-intercalation (SI) in $NbSe_2$ phases where the additional Nb metal atoms lie in octahedral holes between $NbSe_2$ layers. Most of the structures were characterized by X-ray powder diffraction method, and very few investigations using single crystal X-ray diffraction technique were performed.

The Nb-Se phase diagram has been established on the basis of the homogeneity range of several polytypes which mostly depend on the method and the synthesis conditions (*Predel, 1997*). However, and according to the recent findings (*Ivanova et al., 2019*), the diagram may contain some ambiguous data and should be revisited

The selenide $2H$-$NbSe_2$ revisited by single crystals (*Meerschaut & Deudon, 2001*) and the selenide $3R$-$NbSe_2$ investigated on the basis of the precession photograph (*Huisman, Kadijk & Jellinek, 1970*) have been both found stoichiometric; while the sulfur polytype $3R$-$Nb_{1+x}S_2$ ($x = 0.06$ and $0.09$) (*Meerschaut & Deudon, 2001*; *Powell & Jacobson, 1981*) is non-stoichiometric.

According to *Fisher & Sienko (1980)*, these compounds are non-stoichiometric. Indeed, in the $2H$-$NbSe_2$, the niobium atoms are stacked directly one above the other along the c axis, which is not the case for $3R$-$NbSe_2$. In the case of an excessive intercalate niobium a transition from 2H to 3R polytype should then occur in order to minimize the $Nb_{layer}$—$Nb_{extra}$—$Nb_{extra}$ repulsions between these adjacent slabs. This transition was explained by an $NbS_2$-layers rotation mechanism involving stacking faults probabilities between 15–18% (*Katzke, 2002*; *Leroux et al., 2018*).

In this work, three polytypes in the Nb-Se system: $2H$-$Nb_{1.031}Se_2$, $3R$-$Nb_{1.071}Se_2$ and $3R$-$Nb_{1.085}Se_2$ were synthesized by CVT and investigated by single crystal X-ray diffraction.

In particular, the role of vacancy and Nb-Nb interactions on the non-stoichiometry and stacking mode in $NbSe_2$ has been examined.

Both 2H and 3R polytypes may co-exist in the range $0<x<0.07$ and above this limit where only the form 3R predominates as a transition 2H-3R will take place.

Furthermore, by using DFT with CASTEP code, a comparative study involving thermodynamic polytype stability of 2H and $3R$-$Nb_{1+x}Se_2$ ($x = 0, 0.1$) has been attempted in order to support the X-ray diffraction conclusions.

## MATERIALS & METHODS

### Crystal growth and chemical analysis

Single crystals of 2H and 3R-$Nb_{1+x}Se_2$ were obtained by chemical vapor transport (CVT) method during our attempts to prepare ternary $HgNbSe_2$ and $SnNbSe_2$. A stoichiometric amount of pure elements with an excess of Se were mixed and sealed in an evacuated quartz tube (length $\sim$18 cm) using iodine ($<$5 mg/cm$^3$) as a transport agent to favour crystallization. The mixtures (charge $\sim$1g) were placed into a zone tube furnace. The temperature was first increased slowly at 500 °C and held then for 48 h in order to avoid thermal runaway caused by the exothermic reaction between Nb and Se. After that the temperature is turned up between 760−1,050 °C for 15 days. Finally the furnace was allowed to cool slowly to room temperature. Crystals of appreciable sizes were grown in the cold end of the tube; 2H+3R crystals are obtained at $\sim$760 °C while the 3R crystals are obtained at $\sim$1,050 °C. The analysis results by X-ray energy dispersive spectroscopy (XEDS) with the TEM (EDX Oxford Instruments) on several different crystallites of each polytype sample gave an approximate atomic ratio of 1:2 for Nb and Se [for a selected *3R*-crystal: Nb (at%) = 29,9 and Se (at%) = 70,1].

### Single crystal structure determination

Single crystals data were recorded on a Kappa Apex II CCD X-ray diffractometer (*Bruker, 2006*) with graphite-monochromated MoK$\alpha$ ($\lambda = 0.71071$ Å) radiation. The reflection intensities were integrated with the SAINT (*Bruker, 2006*); SADABS was used for empirical absorption correction (*Sheldrick, 2002*, and *Petříček, Dušék & Palatinus, 2006*) was performed for the structure refinement.

The crystal structure of *3R*-$Nb_{1.071}Se_2$ was obverse/reverse twinned while the *3R*-$Nb_{1.085}Se_2$ was twinned by inversion; with a refined twin domain fraction of 0.611(5): 0.389(5) and 0.85(8): 0.15(8) respectively. The CIF files containing details of the structure refinements are available in the File S1.

In the final cycles of refinement, all the atoms in the different structures were refined anisotropically.

Details concerning the structure refinement and final results are presented in Table 1 while atomic coordinates, anisotropic displacement parameters and selected bond distances are listed in Tables S1–S3.

### Theoretical calculations

The DFT calculations were performed using the plane-wave pseudopotential method implemented in the Cambridge Sequential Total Energy Package (CASTEP) code (*Clark et al., 2005*) of the Material Studio program from Accelrys (*Materials Studio CASTEP Manual Accelrys 2010, 2010*). The GGA-PBE functional (*Hammer, Hansen & Nørskov, 1999*) was used to model the exchange and correlation interactions. The Broyden–Fletcher–Goldfrab–Shanno (BFGS) method was used to carry out the geometrical optimization (*Shanno, 1985*). The plane-wave cut-off energy was adopted to be 500 eV and the Monkhorst–Pack scheme (*Perdew, Burke & Ernzerhof, 1996*), k-point grid sampling was set to $7 \times 1 \times 1$ in the Brillouin zone. As far as self-consistent condition setting is concerned, the total energy was

**Table 1  Selected single crystal data and structure refinement parameters for the 2H-Nb$_{1.031}$Se$_2$, 3R−Nb$_{1.071}$Se$_2$ and 3R-Nb$_{1.085}$Se$_2$ polytypes.**

| Nb$_{1+x}$Se$_2$ Polytype | 2H-Nb$_{1.031}$Se$_2$ | 3R-Nb$_{1.071}$Se$_2$ | 3R-Nb$_{1.085}$Se$_2$ |
|---|---|---|---|
| Molar mass (g.mol$^{-1}$) | 253.7 | 257.3 | 258.1 |
| Crystal size (mm$^3$) | $0.36 \times 0.16 \times 0.05$ | $0.33 \times 0.31 \times 0.015$ | $0.20 \times 0.19 \times 0.03$ |
| Space group, Z | P6$_3$/mmc, 2 | R3m, 3 | R3m, 3 |
| Unit cell dimensions (Å) | $a = 3.4475(3)$ | $a = 3.4512(4)$ | $a = 3.4670(4)$ |
| | $c = 12.5702(11)$ | $c = 18.827(3)$ | $c = 18.866(2)$ |
| c/a | 3.646 | 5.455 | 5.441 |
| (c/n)/a | 1.823 | 1.818 | 1.836 |
| Volume (Å$^3$) | 129.38(2) | 194.21(4) | 196.39(4) |
| Calculated density (g cm$^{-3}$) | 6.5101 | 6.5987 | 6.5442 |
| Absorption coefficient (mm$^{-1}$) | 32.515 | 32.658 | 32.33 |
| Angular range $\theta$ (°) | 3.24–27.48 | 6.50–39.95 | 3.24–29.86 |
| Index ranges | −3<h<4; −4<k<4 | −6<h<6; −6<k<4 | −4<h<4; −4<k<4 |
| | −16<l<16 | −32<l<33 | −25<l<25 |
| Total recorded reflections | 1568 | 1707 | 947 |
| Independent reflections, Rint | 80, 0.0505 | 547, 0.0498 | 188, 0.054 |
| Reflections with I >3$\sigma$(I) | 75 | 534 | 176 |
| T$_{min}$/T$_{max}$ | 0.2910/0.7456 | 0.000/0.615 | 0.368/0.746 |
| Number parameters | 9 | 14 | 15 |
| R$_1$, wR$_2$ (all) | 0.0882/ 0.1033 | 0.0232/ 0.0271 | 0.057/ 0.084 |
| Flack parameter | | | 0.15(8) |

less than $3.10^{-2}$ eV atom, and the maximum displacement and maximum stress allowed was $10^{-3}$ Å and $5.10^{-2}$ GPa respectively. The valence electronic configurations considered for atomic pseudopotential calculation are Nb: $4s^2 \, 4p^6 \, 4d^4 \, 5s^1$ and Se: $4s^2 \, 4p^4$.

To investigate the relative stability of the Nb-SI in the 2H and 3R polytype with an excess of niobium ($x = 0.1$), $5 \times 1 \times 1$ and $5 \times 2 \times 1$ supercells respectively are adopted, corresponding to the ordered configurations Nb$_{10}$Se$_{20}$ ($x = 0$) and Nb$_{11}$Se$_{20}$ ($x = 0.1$) to simulate both pure and SI systems.

The formation enthalpy $\Delta$E at $T = 0$ K is expressed as the difference in total energy of the supercell calculated by DFT and the chemical potentials E of Nb (cubic Fm-3m) and Se (trigonal P3$_1$21) calculated from their respective bulks in the same computation conditions.

$$\Delta E = \frac{1}{(x+y)} [E_{DFT} - xE_{Nb} - yE_{Se}]$$

where $x$ and $y$, respectively represent the Nb and Se atoms number in the supercell structure model.

The calculated free energy $\Delta$G at typical synthesis temperatures of 760 and 1,050 °C only include configurational entropy due to intercalation, according to (*Ivanova et al., 2019*).

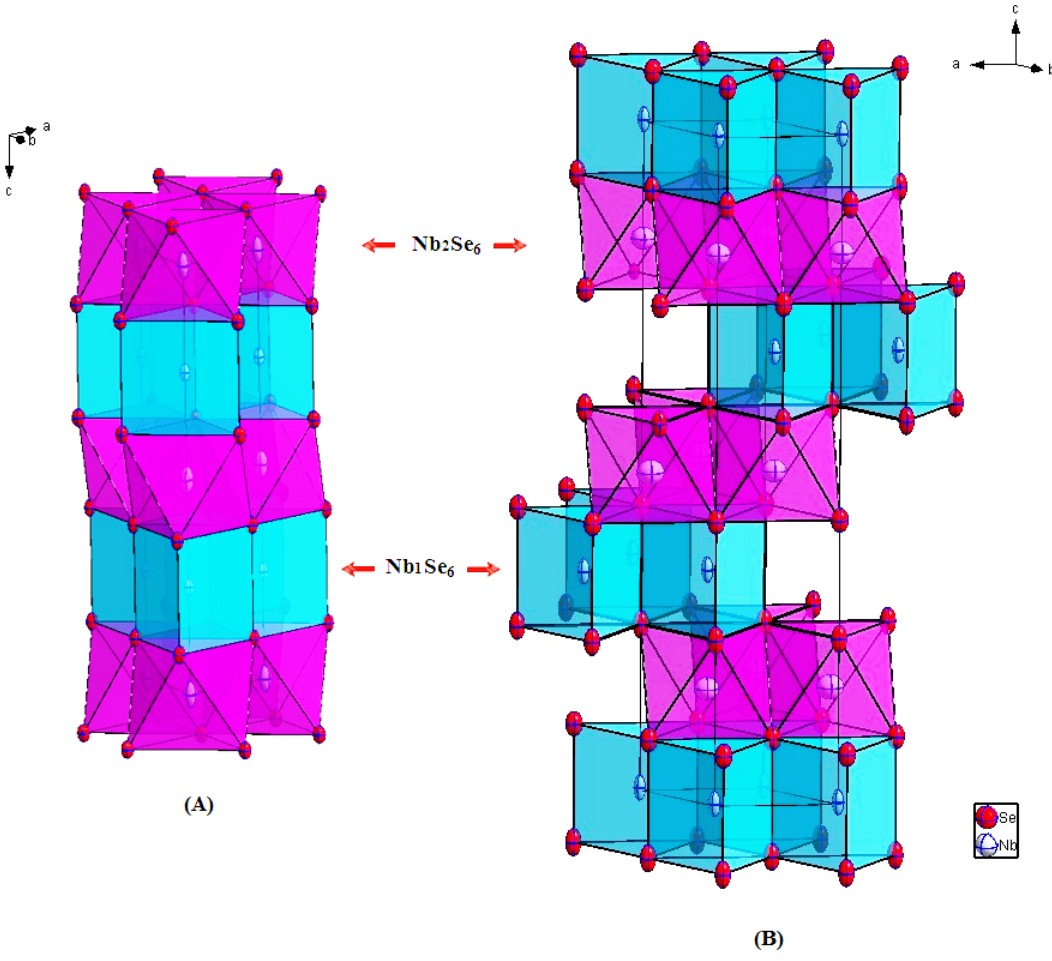

**Figure 1** **Crystal structure of (A) $2H$-$Nb_{1.031}Se_2$; (B) $3R$-$Nb_{1.085}Se_2$ showing the prismatic packing of $Nb1Se_6$ polyhedron (drawn in blue) and $Nb2Se_6$ (drawn in purple).** The extra Nb2 atoms are located into the octahedral sites ($Nb2Se_6$ drawn in purple) within the van der Waals gap.

## RESULTS AND DISCUSSION

### X-ray experimental crystal structure

The polytype 2H-$Nb_{1.031}Se_2$ is almost stoichiometric (Fig. 1A). Additional metal Nb2 atoms with a refined occupancy about 7.3% are located into the octahedral sites within the van der Waals VDW gap with the stacking sequence AcA$\delta$BcB (A,B refer to the selenide layers; c to the completely filled niobium layers and $\delta$ to the partially filled niobium layers).

Polyhedrons around the Nb atoms are not distorted with six equivalent Nb-Se distances. The Nb1- 6Se = 2.597(2) Å around Nb1 (in a trigonal prismatic coordination) are somewhat larger than the Nb2-6Se = 2.477(2) Å, observed around Nb2.

A characteristic feature of this 2H- structure is the short Nb1-Nb2 = 3.1425(6) Å distance, suggesting the formation of strong metal–metal repulsions between the adjacent layers, but can also be analyzed in terms of electrostatic repulsion (Fig. 2A).

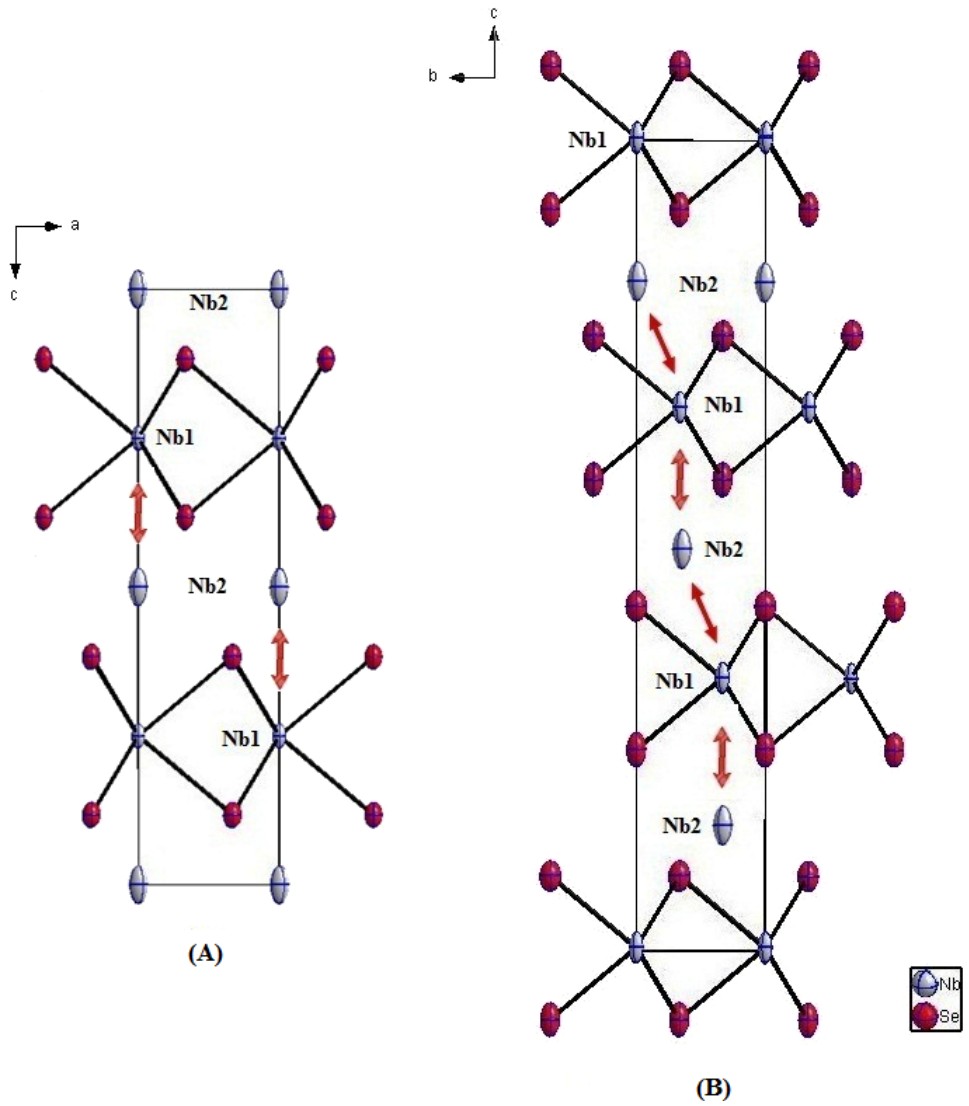

**Figure 2** Projection of the crystal structures of (A) $2H$-$Nb_{1.031}Se_2$; (B) $3R$-$Nb_{1.085}Se_2$ along the [100] direction, showing the $Nb_{layer}$—$Nb_{extra}$—$Nb_{layer}$ repulsions (red arrows). Thermal ellipsoids are drawn at 95% probability displacement level.

The relaxation probably occurs following a defect model by introducing vacancy (about 4%) in the Nb1 filled layers. The Nb1 atoms are then displaced forward the created vacancy to minimize the formation of Nb1-Nb2 pairs interactions (*Tronc & Moret, 1981*; *Amzallag et al., 2007*).

The composition obtained with the structure refinement is: $2H$-$(Nb_{0.96(2)}\ \Delta_{0.04})$ $Nb_{0.073(13)}Se_2$

($\Delta$ denotes Nb vacancy in the fully layers) (ie) $2H$-$Nb_{1.031}Se_2$.

The Nb1-Se distances in the $3R$-$Nb_{1.071}Se_2$ polytype are very comparable [2.594(9) –2.601(8) Å] and close to the values [2.59(8)–2.62(8) Å] reported by *Brown & Beerntsen (1965)*. The refined occupancy (sof) of the additional Nb2 atoms (7.1%) is almost equal

to that observed in the previous 2H structure with the stacking sequence $AbA\delta BcB\alpha CaC\beta$ (A, B, C refer to the selenide layers; a, b, c to the completely filled niobium layers and $\alpha, \beta, \delta$ to the partially filled niobium layers).

The octahedrons $Nb2Se_6$ are slightly distorted with three short [2.288(10) Å] and three long [2.698(14) Å] distances. The Nb2 atoms are then shifted from the center of the octahedrons in order to minimize the Nb1-Nb2 [3.425(17) Å] interactions.

In order to avoid the direct stacking Nb1—Nb2—Nb1, a relaxation occurs following the transition 2H-3R model and consequently the Nb1-Nb2 distance increases to 3.425(17) Å. Indeed, in the 2H structure the relaxation by introducing high concentration of vacancy in the filled Nb layers may destabilize the structure by breaking the Nb-Nb bonding across the face sharing polyhedrons.

In the $3R\text{-}Nb_{1.085}Se_2$ (Fig. 1B), the refinement reveals a high occupancy, about 11.8%, of additional Nb2 atoms. By comparing with the previous 3R structure, the $Nb2Se_6$ octahedrons are less distorted Nb2-Se = 2.362(11)-2.621(14) Å and the Nb1-Nb2 distance (3.36(2) Å) becomes shorter (Fig. 2B). This can be explained by a shrink of the Nb-Nb layers with high content of extra Nb2 atoms.

In this case, both models exhibit relaxation: 2H-3R transition followed by a defect model introducing about 3% of vacancy in the Nb filled layers. A comparable defect was observed in the $3R\text{-}Nb_{1.06}S_2$ (*Powell & Jacobson, 1981*).

The composition obtained with the structure refinement is: $3R\text{-}(Nb_{0.97(2)} \ \Delta_{0.03})Nb_{0.118(13)}Se2$ ($\Delta$ denotes Nb vacancy in the fully layers) (ie) $3R\text{-}Nb_{1.085}Se_2$.

In $Nb_{1+x}Se_2$, as in the layered transition metal dichalcogenides, vacancies in the sublattice of the metal (Nb) and chalcogen (Se) are expected.

It is quite clear that the increase in the additional Nb induces changes in the pressure of saturated Se vapor above the surface of the growing crystal, and then affects the concentration of selenium vacancies. However, a free refinement of the occupancy of Se atoms in the three polytypes leads to a fully occupied sites, possibly due to the excess of Se used as starting material. Therefore, the generation of vacancies in the sublattice of Nb should be correlated to the location of Nb interlayer.

By the presence of extra Nb atoms, a significant residual electron densities are observed in the crystal structure refinements stacking in a way that violates the ideal 2H (ABAB..) and 3R (ABCABC..), about 7.3% for the former and 7.1–11.8% for the latter. The remaining 92.7% and 92.9–88.2% are unfaulted (see CIFs in Electronic Supplemental Files). Certainly, a stacking fault is a common feature in this kind of layered structures, with strong diffuse scattering observed along the $c^*$ axis in electron diffraction patterns (see Fig. S1).

An indication about the relative stability of 2H and $3R\text{-}Nb_{1+x}Se_2$ structures, can be obtained by comparing the variation of c/a and (c/n)/a ratios with x, where a and c are the unit cell parameters and n the number of layers in the stacking sequence; $n = 2$ and 3 for 2H and 3R respectively.

As shown in Fig. 3, this ratio change is relatively small with the increase of x for both polytypes with a discontinuity at the 2H to 3R transition around $x = 0.07$. It is assumed that both polytypes may co-exist in the range $0 < x < 0.07$, and above this limit only the form 3R predominates. Comparable results were observed in the study of the non-stoichiometric

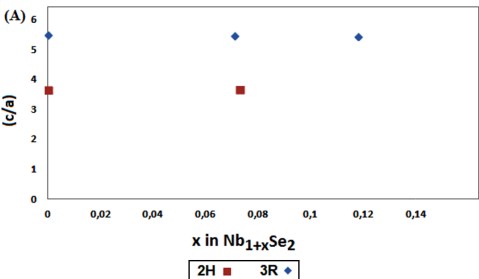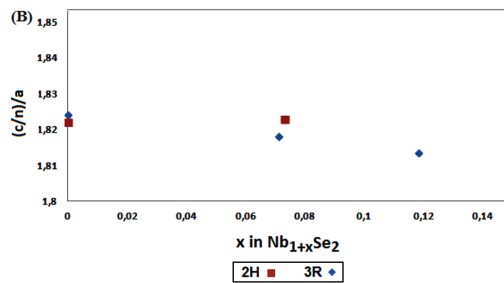

**Figure 3** **The reduced lattice parameters ratio (A) c/a; (B) (c/n)/a for 2H and 3R-$Nb_{1+x}Se_2$ ($0 < x < 0.1$)**
**polytypes.** The values at $x = 0$ for 2H and 3R are calculated respectively from the references (*Brown &*
*Beerntsen, 1965*; *Meerschaut & Deudon, 2001*).

**Table 2** **The optimized structural constants, selected bond lengths, calculated formation energy ΔE (eV) at 0 K and free energy $\Delta G_{1033,1323}$ (eV)**
**at 1033 and 1323 K for both pure and NbSI 2H and 3R-$NbSe_2$ respectively.**

| Polytype | a , c (Å) | c/a | (c/n)/a | Nb1–Se , Nb2–Se (Å) | δ (Å) | ΔE | $\Delta G_{1033,1323}$ |
|---|---|---|---|---|---|---|---|
| 2H-$NbSe_2$ | $a = 3.463$ , $c = 13.210$ | 3.814 | 1.907 | 2.587–2.600 | 0.013 | −1.11 | −0.96 , −0.97 |
| 2H-$Nb_{1.1}Se_2$ | $a = 3.470$ , $c = 13.807$ | 3.979 | 1.990 | 2.569-2.687, 2.564-2.579 | 0.118 , 0.195 | - 0.93 | −0.97 , −0.98 |
| 3R-$NbSe_2$ | $a = 3.506$ , $c = 19.708$ | 5.621 | 1.873 | 2.597–2.600 | 0.03 | −1.25 | |
| 3R-$Nb_{1.1}Se_2$ | $a = 3.412$ , $c = 21.679$ | 6.300 | 2.100 | 2.592–2.697, 2.610–2.797 | 0.105 , 0.187 | −0.94 | |

**Notes.**
Note: Same notation for Nb and Se atoms as used in X-ray refinement tables. The distortion δ (Å) is the 6 difference between the longest and shortest Nb-Se bond distance.

2H and 3R-$Nb_{1+x}S_2$ ($0.07 < x < 0.18$) by powder diffraction (*Fisher & Sienko, 1980*).
The mechanism involving the 2H-3R transition is still unclear, and may be attributed to
the phase limit of the non-stoichiometric $Nb_{1+x}Se_2$. The difference in energy between
polytypes is small, thus the possibility to obtain a mixture of polytypes by changing the
conditions of synthesis cannot be ruled out.

The present results are in agreement with earlier studies by *Huisman, Kadijk & Jellinek*
*(1970)* and *Selte, Bjerkelund & Kjekshus (1966)*. The mixture 2H and 3R polytype were
prepared at temperature of 760 °C in the range $0.07 < x < 0.118$, yet the 3R phase at $x = 0.07$
required higher temperature $1,050−1100$ °C. Further experimental and theoretical works
are mandatory to determine the x range of 2H-3R transition accurately.

## Theoretical results

To corroborate the single X-ray conclusions, the stability of four structures: the pure 2H-
$NbSe_2$ and 3R- $NbSe_2$, 2H-$Nb_{1.1}Se_2$ and 3R-$Nb_{1.1}Se_2$ with 10% of extra Nb-SI in the pure
2H and 3R, respectively, have been investigated. The optimized structural parameters of
the supercell structures are summarized in Table 2.

The calculated c lattice parameters of 2H and 3R-$NbSe_2$ are slightly overestimated
compared to the experimental values by 5.28% and 4.38% respectively. This discrepancy
is due to the failure of GGA (local and semi-local approximations for the exchange–
correlation) in describing the van der Waals interactions.

Structural geometrical optimization revealed that additional Nb2 atoms incorporated in 2H and 3R-Nb$_{1+x}$Se$_2$ with $x = 0.1$, obviously increase the c parameter by 7.61% and 9.73% respectively while the a parameter remains almost constant.

The Nb1-Se distance slightly increases in pure 2H and 3R-NbSe$_2$ (2.587–2.600 Å and 2.597–2.600 Å) compared to 2H and 3R-Nb$_{1.1}$Se$_2$ (2.569 –2.687 Å and 2.592–2.697 Å) while the Nb2-Se distances range from 2.564–2.759 Å and 2.610–2.797 Å respectively.

The calculated formation energies at $T = 0$ K (Table 2) indicate that the four structures polytypes can be thermodynamically stable. However, due to the strong Nb-Nb interactions the Nb ($x = 0.1$) SI system is found to be the least favorable. The close free energy ΔG values at high temperature suggest that facile phase transition between the two polytypes may occur, as established between Nb$_2$Se$_3$ and Nb$_{1.33}$Se$_2$ polytypes *Ivanova et al. (2019)*. These results can explain the experimental stability of both 2H and 3R between $0 < x < 0.07$, and confirm the hypothesis of a phase transition from 2H to 3R beyond $x = 0.1$.

The band structures and density of states (PDOS) of the foregoing four cases have been presented to study the impact of the Nb-SI on the electronic structure for both 2H and 3R-NbSe$_2$.. Figures 4A and 4B show the band structure of the pure 2H and 3R respectively. The presence of several bands across the Fermi level E$_F$ reveals the metallic nature of the pure polytypes.

Partial density of states (PDOS) (see Figs. S2A and S2C), indicates that the major contribution in DOS at E$_F$ comes from the hybridization between the Nb-4d and Se-4p orbitals which is responsible for the covalent Nb-Se bonds.

For both 2H and 3R Nb-SI, the number of electronic bands around E$_F$ apparently increases compared to the pure 2H and 3R (Figs. 4C and 4D), and consequently the gap around 2 eV in the Conduction Band disappears. Moreover, the PDOS at E$_F$ decreases and upshifts (see Figs. S2B and S2D) compared to the pure polytypes which implies that a large degree of electrons is transferred by Nb extra atoms into these pure polytypes.

It is worth noticing, that the PDOS of Nb-4d is rather broad and almost located at the same energy with Se-4p, suggesting strong interactions. These phenomenon have been observed in some doped and intercalated 2H-NbSe$_2$ (*Chen et al., 2014*; *Hongping et al., 2014a*; *Hongping et al., 2014b*; *Kouarta, Zanat & Belkhir, 2019*; *Pervin et al., 2020*; *Xiao-Chen et al., 2020*).

## CONCLUSIONS

In conclusion, the non-stoichiometric polytypes 2H-Nb$_{1.031}$Se$_2$, 3R-Nb$_{1.071}$Se$_2$ and 3R-Nb$_{1.085}$Se$_2$ have been successfully synthesized and investigated by single crystal X-ray diffraction. Although the form 3R predominates for values of x greater than 0.07, both 2H and 3R polytypes may co-exist in the range $0 < x < 0.07$. A transition 2H to 3R polytypes, followed by model vacancies in the host Nb sublattice should take place in order to minimize the $Nb_{layer}$—$Nb_{extra}$—$Nb_{layer}$ repulsions between these adjacent slabs.

The calculated formation energies of 2H and 3R-Nb$_{1+x}$Se$_2$ ($x = 0, 0.1$) by DFT, indicate that both pure and Nb-SI systems can be thermodynamically stable, and suggest an easy phase transition between polytypes. The theoretical outcomes reveal the metallic nature of

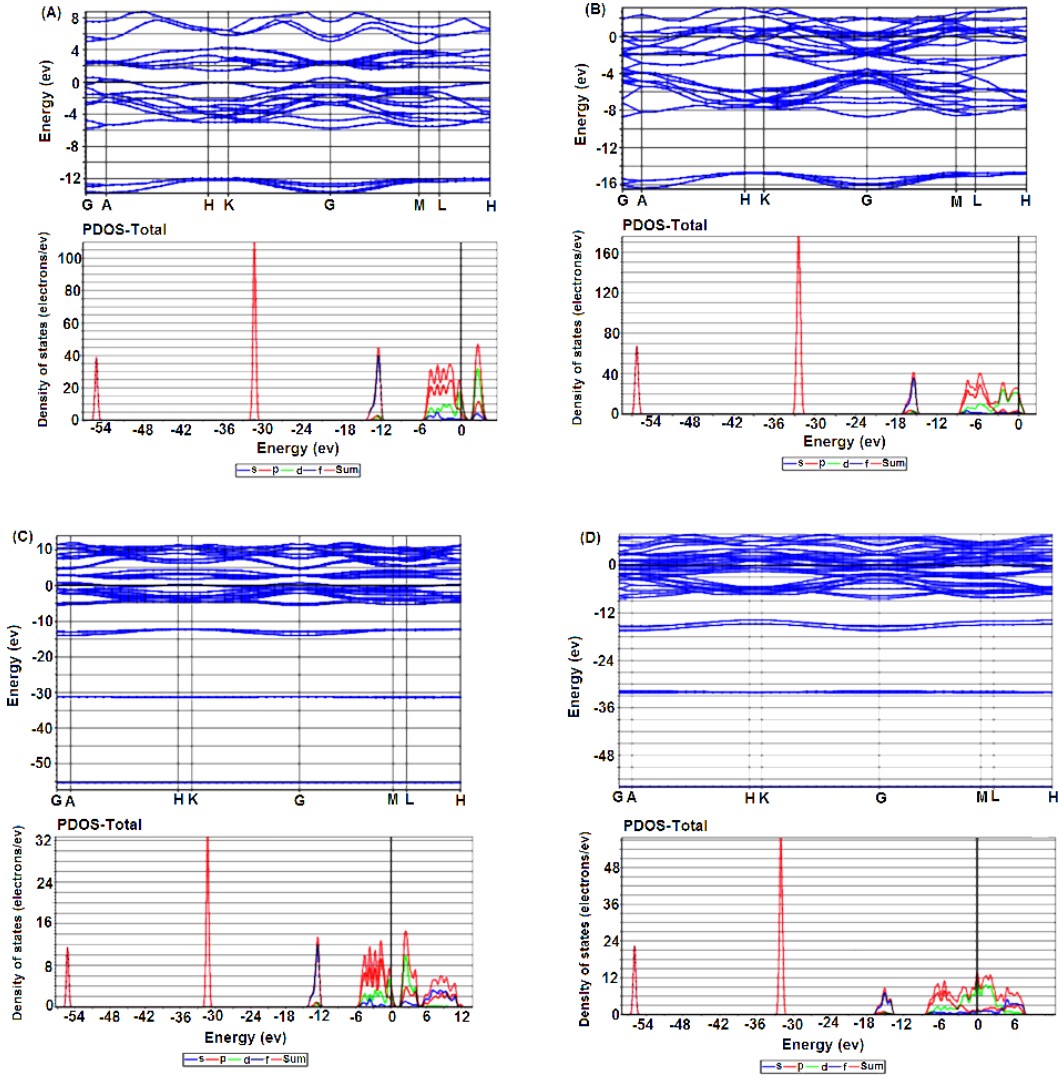

**Figure 4** Electronic band structure and their corresponding DOS of (A) 2H-NbSe$_2$; (B) 2H-Nb$_{1.1}$Se$_2$, (C) 3R-NbSe$_2$; (D) 3R-Nb$_{1.1}$Se$_2$.

these polytypes with an overwhelming number of electrons transferred by Nb extra atoms into pure polytypes.

Further work involving experimental and theoretical investigations on the Nb-Se system is needed to elucidate the 2H-3R transition mechanism.

### Funding

The authors received no funding for this work.

## Competing Interests

The authors declare there are no competing interests.

## Author Contributions

- Mohamed Sidoumou conceived and designed the experiments, performed the experiments, performed the computation work, authored or reviewed drafts of the paper, and approved the final draft.
- Soumia Merazka and Mohammed Kars conceived and designed the experiments, performed the experiments, analyzed the data, performed the computation work, prepared figures and/or tables, authored or reviewed drafts of the paper, and approved the final draft.
- Adrian Gómez-Herrero and Roisnel Thierry conceived and designed the experiments, performed the experiments, analyzed the data, performed the computation work, authored or reviewed drafts of the paper, and approved the final draft.

## Data Availability

Supplementary crystallographic data are available at the joint CCDC/FIZ Karlsruhe deposition service: https://www.ccdc.cam.ac.uk/structures/search?pid=ccdc:1976959, 1976960,2004939.

Data are available in the Supplementary Files.

## Supplemental Information

Supplemental information for this article can be found online at http://dx.doi.org/10.7717/ peerj-ichem.2#supplemental-information.

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
