# Peer review of "X-ray diffraction and theoretical study of the transition 2H-3R polytypes in $Nb_{1+x}Se_2$ $(0 < x < 0.1)$"

_PeerJ Inorganic Chemistry, doi:10.7717/peerj-ichem.2_

## Round 0.1 · original submission · Major Revisions

Based on the reviewers comments, the paper needs major revisions and resubmission prior to its acceptance. Please address the reviewers questions point by point and submit a revised version of the manuscript within 55 days. If you need to extend the deadline, please get in contact with us.

Best regards,

Jordi Cirera

Reviewer 1 ·

Basic reporting

no comment

Experimental design

no comment'

Validity of the findings

no comment'

Additional comments

Title: X-ray diffraction and theoretical study of the transition 2H-3R polytypes in Nb1+xSe2 (0 < x <0.1)

The main conclusion for this paper is different structure switching between 2H to 3R by self-intercalation in the niobium diselenide system. Although this behavior is not new, but the data is useful for further study the polymorphism in the TMDs. Therefore, I suggest it can be accepted to be published in PeeJChemistry Journals after major revisions with answering the following questions:

Some issues to be address:

1. Introduction, it is well-known that 2H-NbSe2 is one of the most famous TMDs, where charge density wave (CDW) in coexistence with superconductivity. And the superconducting transition temperature is around 7.2 K which is the highest Tc among the pristine TMDs. However, the authors state that "The system NbSe2 has been the subject of many investigations, since it exhibits incommensurate charge density waves (CDW) and superconductivity phenomenon below to 4K (Wilson, Disalvo & Mahajan, 1975)." is obviously wrong. Evidently, the superconductivity can be seen above 4 K in 2H-NbSe2.

2. Introduction, up to now, there are innumerable reports on polymorphism in the TMDs by tuning systhesis temperature (such as MoTe2, from ref. 1966 Acta. Cryst. 20 268.), chemical doping (such as Mo1-xNbxTe2 from Ref. 2015, APL Materials 3(4); TaSe2-xTex from Ref. Proc Natl Acad Sci USA, 2015, 112 E1174) or self-intercalation (such as TaxSe2) and so on. The authors should mention these case and make the readers easily understanding.

3. There is a lot mistakes and typos in the whole manuscript. The authors should go through it and fix them, such as "formulaeTX2" in the introduction, 7x1x1 in line 131, Nb2Se6 in 173 and so on.

4. The authors should perform the resistivity and magnetic susceptibility measurements to check the CDW and superconductivity on these 2H and 3R-Nb1-xSe2 samples, compared with that of the 2H-NbSe2.

Annotated reviews are not available for download in order to protect the identity of reviewers who chose to remain anonymous.

Reviewer 2 ·

Basic reporting

Numerous language deficiencies are present, (e.g. out-of-place commas, adjectives without corresponding nouns or missing verbs, such as "Both have been found stoichiometric;", etc.) .

Experimental design

Methods should be explained in more detail. For example, authors state: "The mixture was heated between 760-1050 C for 15 days" Does this mean that the temperature was changed from 760 to 1050 C over a span of 15 days, that several synthesis were performed at different temperatures (each constant during 15 days), or that the temperature was allowed to fluctuate between 760-1050 during that time? Please clarify. I also cannot tell which conditions gave rise to each of the polytypes, or whether a mixture wwas obtained and then the specific crystals were picked individually for further study.

" ...then slowly cooled to room temperature. Crystals of appreciable sizes were grown in the cold end of the tube. " This seems to imply that the cooling was not homogeneous, and a description of the non-homogeneity of the cooling process is therefore needed.

" an approximate ratio of 1:2 for Nb and Se [for 3R-Nb1.085Se2: Nb (at%) = 29,9 and Se (at%) = 70,1]" Authors should clarify whether the stated ratio is mass or molar ratio. If those are molar ratios, that translates to a 0.85:2 Nb/Se ratio. If those are mass ratios, they translate to a 0.725:2 Nb/Se ratio. In any case those are quite different from the claimed 1.085:2 Nb/Se ratio.

Some extra detail is needed for the theoretical computations: please provide (as Supporting information) the DFT energies of all systems, as well as their geometries. I also notice that the formation enthalpy you compute is not the actual enthalpy, but the energy divided by the number of atoms. Why? Also, at what temperature was enthalpy computed? Authors should state that. Why was enthalpy selected rather than free energy? Doesn't CASTEP allow the computation of hessians and vibrational/rotational contributions, ZPE, etc. , and hence the entropy
and Cv/Cp values needed to compute G?

Validity of the findings

See below

Additional comments

in line 56 " since it exhibits [...] superconductivity phenomenon below to 4K (Wilson, Disalvo & Mahajan, 1975)" I guess authors mean "below 4 K", but the quoted reference shows Tc as 7.3 K (in its table 5, page 189)

line 74 "revisited by single crystals ". I guess authors mean "revisited by single crystal X-ray diffraction"

line 80 "By the fact of an excess of niobium, a transition 2H to 3R polytype should occur in order to minimize the Nblayer—Nbextra—Nbextra repulsions between these adjacent slabs. " I am afraid that the language here is again cryptic: aren't both the 2H and 3R polytypes NbSe2? Do authors mean instead that as additional (supra-stoichoiometric) Nb is added the 3R polytype becomes energetically favored relative to the 2H?

lines 193-195 "it is quite clear, that the increase in the additional Nb , induces changes in the pressure of saturated Se vapor above the surface of the growing crystal, and then affects the concentration of selenium vacancies." Authors may suspect that, but I am afraid I do not think that is at all clear: did the authors monitor the Se vapor profile above the crystals duting growth? (Is that even technically possible?) I may be misreading, bu it seems to me that authors have only mentioned Nb vacancies before this sentence, and therefore I do not see how they can posit an explanation of the Se vacancies that (apparently) are not reported to be present (an on the next paragraph are stated to not exist "A free refinement of the occupancy of Se atoms in the three polytypes, leads to a fully occupied sites")

Figure 3 is not immediately readable, as the reader cannot tell whether the points at the upper right of each graph are calculated/experimental points or are a legend instead. Please correct that. I also fail to see a discontinuity around x=0.07, since (apart from the point that may (or not) be a legend, I see no other 2H points, and the 3R points are not discontinuous).

The legends in Figure 4 are barely readable. please correct.

The discussion of the computed stabilities is deficient: from the data in table 2, I compute formation enthalpies also of - 66.5 kcal/mol for 2H-Nb1.1Se2 and -67.2 kcal/mol for 3R-Nb1.1Se2 (indeed indistiguishable with the computational methods used). The difference between the stoichoiometric compounds is much larger -76.8 kcal/mol for 2H-NbSe2 and 86.5 kcal/mol for 3R-NbSe2. Interesting as enthalpy differences are, stability cannot be directly inferred from that , but requires the computation of free energies.

line 252 "Moreover, the PDOS at EF decreases and upshifts (see Figs. S2B and S2D) compared to the pure polytypes, which implies that a large degree of electrons is transferred by Nb extra atoms into these pure polytypes" This should be explained more fully (o at least a citation for a reference detailing how the DOS around Fermi level E(F) relates to the metallic/chalcogenide electron distribution).

There is an important typo in table 2: authors wrote Nb1.01Se2 , where they really should have written Nb1.1

---

## Round 0.2 · Minor Revisions

Dear Prof. Kars,

Your paper has been reevaluated by the referees, and it requires minor revisions. Please address such comments prior to the acceptance of your paper for publication in PeerJ.

Best regards,

Reviewer 1 ·

Basic reporting

It is clear than before.

Experimental design

The data is enough for published.

Validity of the findings

It is meet the standards.

Annotated reviews are not available for download in order to protect the identity of reviewers who chose to remain anonymous.

Reviewer 2 ·

Basic reporting

see below

Experimental design

see below

Validity of the findings

see below

Additional comments

I am generally satisfied with the authors responses. There are still a few issues to address:
A) I would have preferred the authors to explicitly state that their computations of free energy (performed using the methodology of Ivanova et al. 2019) only include configurational entropy due to intercalation .

B) I still do not understand how the Nb1.085:Se2 sample could afford a 29.9%Nb - 70.1 % Se mass (or molar) ratio. That should be clarified.

C) In line 224 authors still state " As shown in Fig. 3, this ratio change is relatively small with the increase of x for both polytypes with a discontinuity around x = 0.07. " but no discontinuity can be seen. Please consider rephrasing this, for clarification of what you mean.

---

## Round 0.3 · accepted · Accept

Dear Dr. Kars,

It is my pleasure to inform you that your paper is now accepted for publication. Congratulations, and we hope that you will be sending more excellent work for publication in the PeerJ journals.

Congratulations.

Best,

Jordi